# Preparation of Organic-Inorganic Coupling Phase Change Materials with Enhanced Thermal Storage Performance via Emulsion Polymerization

**DOI:** 10.3390/ma15093373

**Published:** 2022-05-08

**Authors:** Xifeng Lv, Xuehua Shen, Luxiang Zhang, Yazhou Wang, Fang Wang

**Affiliations:** 1College of Chemistry and Chemical Engineering, Tarim University, Alar 843300, China; rainspirit_shen@163.com (X.S.); zlx0908zlx@163.com (L.Z.); w15719632978@163.com (Y.W.); 2State Key Laboratory of Organic-Inorganic Composites, Beijing 100029, China; 3School of Environmental Engineering and Chemistry, Luoyang Institute of Science and Technology, Luoyang 471023, China; 4Henan Province International Joint Laboratory of Materials for Solar Energy Conversion and Lithium Sodium Based Battery, Luoyang 471023, China

**Keywords:** phase change heat storage, emulsion polymerization, cetyl alcohol, diatomite, organic-inorganic coupling

## Abstract

The serious phase separation in inorganic phase change materials, and easy leakage of organic phase change materials are the main obstacles to the practical batch application of phase change heat storage materials. To solve these problems, in this work, emulsion polymerization is introduced as the method for preparing organic-inorganic coupling phase change material (oic-PCM) with high heat storage performance using polyacrylamide (PAM) as the wall material and organic phase change material of cetyl alcohol as the core material, and diatomite is used as a supporting substrate to absorb inorganic sodium sulfate decahydrate (SSD). A differential scanning calorimeter (DSC), X-ray diffractometer (XRD), dust morphology and dispersion analyzer, and thermal conductivity tester were used to characterize the prepared organic-inorganic coupled phase change materials and investigate their performance. The research results show that when the mass fraction of cetyl alcohol is 68.97%, the mass fraction of emulsifier is 3.38%, and the mass fraction of sodium sulfate decahydrate/diatomite is 3.40%. The phase change latent heat of the organic-inorganic coupled phase change material is as high as 164.13 J/g, and the thermal conductivity reaches up to 0.2061 W/(m·k), which proves that the prepared organic-inorganic coupled phase change material has good heat storage performance, showing its good application prospects.

## 1. Introduction

Due to the impact of energy shortages, energy conservation has gradually become the focus of attention. Among all the energy consumed by human beings, the energy consumption of buildings occupies a considerable proportion [1]. Therefore, it is of great significance to develop a material that can reduce the energy consumption of buildings. As a kind of energy-saving material, phase-change energy storage materials have received widespread attention because they can store and release heat during the phase-change process, effectively reduce the energy consumption of buildings, and thus improve indoor comfort [2,3,4].

The phase change material (PCM) can solitarily change the physical state in the constant temperature while its chemical properties remain unchanged. According to the phase change temperature, PCM can be divided into low temperature phase change materials, medium temperature phase change materials, and high temperature phase change materials [5]. Phase change materials suitable for use in the field of building energy storage often require a suitable phase change temperature and are usually linked to medium temperature phase change materials. Commonly, phase change energy storage materials can be divided into two categories according to their chemical composition, organic and inorganic phase change materials, both of which have their own advantages [6]. Organic phase change materials have the advantages of stability and no overcooling that inorganic phase change materials do not have, but there are phenomena, such as lax packaging and liquid leakage, constantly accompanied by the phase change [7,8,9,10]. To this end, organic-inorganic phase change materials are expected to solve the above mentioned problems and achieve synergistic effects from both of organic and inorganic materials [11,12].

Recently, some methods have been developed for using composite materials to overcome the disadvantages of inorganic and organic phase materials, such as encapsulation, impregnation, and microcapsule packing, as to enhance the phase change heat storage properties of materials [13,14,15]. The composite PCMs primarily refer to the complex between two or more materials that can overcome the performance deficiencies of a single material, and often give some new excellent properties to the material [16]. The method of encapsulation can achieve a larger heat transfer area and reduce the possibility of the PCM reacting with the environment. Moreover, this technique ensures the decreasing of subcooling and controlling the volume change of the storage materials during the phase transition. Different materials are being employed to encapsulate PCMs, e.g., acrylics, urea, formaldehyde, and silica-based polymers, as well as metal- and carbon-based composites such as graphene and graphite, among others. The application of these encapsulating methods allows to improve the thermophysical performance of PCM. This has already been demonstrated for these composite PCMs [17,18,19,20]. Nevertheless, the information available on the encapsulation of inorganic PCMs is limited and not integrated in a single source. The impregnation is usually used to realize the coating of inorganic salt phase change materials. For example, Deng et al. [21] reported a form-stable composite phase change material using expanded vermiculite (EVM) loaded Na_2_HPO_4_·12H_2_O-alumina via impregnation. The prepared composite material suppresses the degree of supercooling of the inorganic hydrated salt to a certain extent and improves the heat transfer. Furthermore, several studies were carried out on MePCMs (microencapsulated phase change materials) with polymer/inorganic hybrids. It was verified that the incorporation of rigid nanoscale particles into a polymer shell can endow MePCMs with high thermal conductivity and good mechanic toughness. As seen, some great progress and achievements have been made in the research of organic and inorganic phase change materials [22,23,24,25,26,27]. However, these porous materials are generally physically impregnated to encapsulate PCMs in which the organic phase leakage problem has not been solved well. It remains still challenging but it is anticipated to effectively encapsulate PCMs using chemical methods.

Inorganic mineral materials with uniformly distributed micropores, high porosity, developed specific surface area, and micro-void structure, which can be used as a storage space for phase change materials and moisture, are widely used in the field of environmental protection and functional materials [28]. Diatomite is a kind of biochemical sedimentary rock with porous structure whose main component is silica, which can be used to support inorganic phase change materials [29]. Fu et al. [30] prepared a shape-stabilized phase change composite material with a latent heat of 57.1 kJ/kg using stearic acid and diatomite as raw materials by impregnation method. Glauber’s salt, Na_2_SO_4_∙10H_2_O, has abundant sources, the phase transition temperature is 32.4 °C, the latent heat of phase change is 254 J/g, and it can be used for normal temperature phase change materials [31]. Among the numerous organic phase change materials, cetyl alcohol, as a fatty alcohol, shows excellent heat storage performance and its phase change temperature is just suitable for building energy storage [32].

Based on the above considerations, in this paper, an inorganic phase change material of sodium sulfate decahydrate was loaded with porous diatomite. The composite material of sodium sulfate decahydrate/diatomite (SSD/DIAT) and cetyl alcohol are coated in polyacrylamide by emulsion polymerization to make s organic-inorganic coupling phase change material so as to alleviate the supercooling, phase separation of inorganic phase, and the leakage of organic phase. In the coupling materials, porous diatomite serves as support to load the inorganic phase change material of SSD. The composite structure can change the phase change process of SSD by means of the porous structure from the diatomite to achieve spatial confinement, thereby inhibiting the degree of supercooling and the generation of phase separation. While the method of emulsion polymerization provides the possibility to couple the organic cetyl alcohol and inorganic phase by embedding them into the body of PAM porous material, preparing an organic-inorganic composite PCM. The addition of inorganic SSD/DIAT effectively reduces the leakage of cetyl alcohol in the composite PCM and greatly improves the utilization efficiency of cetyl alcohol. Physical characterization and performance testing were carried out to deeply analyze and study the microstructure and heat storage performance of this well-designed organic-inorganic coupled phase change material.

## 2. Experimental

### 2.1. Materials and Chemicals

Acylamide (AM, AR) and N,N’-methylene-bisacrylamide (MBAM, AR) were bought from Macklin (Shangha, China). Polyvinylpyrrolidone (PVP, AR) and cetyl alcohol (C_13_H_34_O, AR) was purchased from Tianjin Fuchen Company (Tianjin, China). Sodium sulfate decahydrate (Na_2_SO_4_·10H_2_O, AR) and ammonium persulfate (APS, AR) were obtained from Aladdin. Tween 85 (AR) was supplied by Shanghai Yuanye Company (Shanghai, China). N,N,N’,N’-tetramethyl ethylenediamine (TMEDA, AR) was purchased from Adamas. Diatomite was offered by Jilin Yuantong Mining Company (Linjiang, China).

#### 2.1.1. Preparation of Sodium Sulfate Decahydrate/Diatomite Composite Material

A certain amount of sodium sulfate decahydrate (SSD) and dried diatomite were weighed at the mass ratio of 4:1. The two materials were then mixed evenly and placed into the vacuum heating oven, which was set to the temperature of 35 °C. The inorganic hydrated sodium sulfate decahydrate was thereafter adsorbed into the porous diatomaceous earth by the action of the vacuum impregnation and physical adsorption. The prepared sodium sulfate decahydrate/diatomite composite phase change material was then put into an oven at 35 °C for 6~8 h. Sodium sulfate decahydrate/diatomite composite material was obtained after heating, grinding, and sieving to be used.

#### 2.1.2. Preparation of Organic-Inorganic Coupling Phase Change Materials

A certain amount of acrylamide (polymerization monomer), N,N-methylenebisacrylamide (crosslinking agent), and polyvinylpyrrolidone (co-emulsifier) were firstly weighed and put into a beaker. Then, 5 mL of distilled water was added to dissolve acrylamide with ultrasound into the water phase for use. Then, the beaker containing the aqueous acrylamide solution was taken out and placed in a water bath with stirring under the temperature of 55 °C. Next, the oil-in-water emulsifier, Tween 85, is joined into the water phase. After 10 min stirring, the molten cetyl alcohol was slowly and uniformly added into the water phase to make a cetyl alcohol-in-water emulsion with the constant stirring and the joint action of the emulsifier and the co-emulsifier. Subsequently, the sodium sulfate decahydrate/diatomite composite material was poured into the emulsion. After stirring uniformly, ammonium persulfate (oxidant agent) and N,N,N,N-tetramethylethylenediamine (reducing agent) were quickly put into the system to make an organic-inorganic coupling phase change material by polymerization and curing. Finally, the material was taken out and dried in an oven at the temperature of 30 °C. The preparation process is shown in Figure 1.

### 2.2. Structural Characterization and Performance Testing

The thermal storage performance of the materials was tested on a differential scanning calorimeter (TA DSC250, TA Instrument Company, New Castle, DE, USA) to scan DSC curves. The test temperature range is controlled at 0~70 °C, the temperature rise and fall rate is 5 °C/min, and the nitrogen served as the protective gas with a flow rate of 210 mL/min. The morphologies and microstructures of samples were characterized by field-emission scanning electron microscope (SEM) (Gemini 300, ZEISS International, Oberkochen, Germany). The particle size of the materials was analyzed using the dust morphology and dispersion analyzer (Rise-3002, Jinan Runzhi Technology Co., Ltd., Jinan, China) with the total magnification of 4×/0.10 or 10×/0.25. The composition of the material was characterized with an X-ray diffractometer (D8 ADVANCE, German Bruker AXS Co., Ltd., Karlsruhe, Germany) at the voltage of 40 kV, current of 40 mA, and the power of 1600 W. The scanning scope is 2θ = 5–90° and the scanning rate is 3°/min. The thermal conductivity was determined by using a thermal conductivity tester (DRL-III, Jinan Runzhi Technology Co., Ltd., Jinan, China) at the conditions with the hot surface temperature of 40.97 °C, the cold surface temperature of 20.01 °C, and the test pressure of 75.4 kPa. The detection of material leakage was carried out in an oven by the sequential implementation of putting the material in an oven at 45–65 °C, taking samples every 10 °C, and heating each sample for 30 min. The thermal stability of the material was investigated using the thermogravimetric analyzer (TA 449 F3, Netzsch Scientific Instruments Trading Co., Ltd., Selb, Germany) at a temperature range of 25~700 °C, a heating rate of 10 °C/min, and with nitrogen as a protective gas.

## 3. Results and Discussion

### 3.1. Optimization of the Preparation Conditions for Oic-PCM

In order to explore the optimal conditions for preparing organic-inorganic coupled phase change materials (oic-PCM), the effects of different mass fractions of cetyl alcohol on the performance of organic-inorganic coupled phase change materials were firstly tested and analyzed. Multiple parallel experiments were implemented by changing the mass fraction of the organic phase change material cetyl alcohol with the fixed emulsifier mass of 0.300 g, and the dosage of sodium sulfate decahydrate/diatomite in composite materials is of fixed value with 0.300 g. A differential scanning calorimeter was used to characterize the as-prepared materials and the scanning DSC curves are shown in Figure 1a. There are two peaks on each DSC curve in the figure. The lower peak temperature of phase change material is 45 °C, and the higher peak temperature of phase change is 50 °C. The former is a phase transition peak from the inorganic phase change material of sodium sulfate decahydrate, and the latter is from the organic phase change material of cetyl alcohol. The area integration of the DSC curve in Figure 1 is carried out to obtain the latent heat of phase change of the organic-inorganic phase change material prepared with different mass fractions of cetyl alcohol. The result of the obtained phase change latent heat is shown in Figure 1b. The change trend of the phase change latent heat shows a trend of first increasing and then decreasing as the mass fraction of cetyl alcohol in the organic phase change material increases within a certain mass fraction. From the parabola of phase transition enthalpy, it is visible that the highest point occurs when the mass percentage of cetyl alcohol in organic phase change material is about 68.97% while the latent heat of phase change is 164.13 J/g. Currently, the latent heat of phase change of the organic-inorganic coupling phase change material is the highest among the prepared materials in this experiment.

In the composite phase change material, polyacrylamide plays a role in coating cetyl alcohol and sodium sulfate decahydrate/diatomite. Therefore, the phase change latent heat of the coupled phase change material of this product depends on the mass fraction of the coated cetyl alcohol and sodium sulfate decahydrate/diatomite in the composite material. The coating rate *R* of the organic-inorganic coupling phase change material is calculated for evaluating the performance level according to the formula as follows [33].
(1)$$R=\frac{{\Delta H}_{mMPCMs}}{{\Delta H}_{mPCMs}}\times 100\%$$

In Formula (1), $\Delta H_{mMPCMs}$ represents the melting enthalpy of the organic-inorganic coupling material (unit: J/g) and $\Delta H_{mPCMs}$ stands for the melting enthalpy of a pure substance (unit: J/g).

Using the melting enthalpy of the obtained product, the coating rates of the cetyl alcohol and sodium sulfate decahydrate/diatomite organic in the composite phase change material were calculated according to the above formula. The coating rate data of the cetyl alcohol and sodium sulfate decahydrate/diatomite composite material for the above-mentioned phase change materials are shown in Figure 2a. It can be found that organic-inorganic coupling phase change material prepared by polyacrylamide coated with cetyl alcohol and sodium sulfate decahydrate/diatomite composite delivers the highest coverage rate up to 67.58% when the added mass fraction of cetyl alcohol is 68.97%. Within a certain mass fraction, the coating rate of the organic-inorganic coupling phase change material goes up with the increase of the cetyl alcohol mass fraction in the organic phase change materials. After the coverage rate reaches its peak, the value begins to show a decrease. As the added mass fraction of cetyl alcohol changes in the organic phase change material, the phase change material coated with polyacrylamide gradually reaches a saturated state. The coating effect of the material becomes worse with the continued addition of a certain mass fraction of cetyl alcohol in the phase change material. The main cause of this phenomenon is that, when the dosages of cetyl alcohol increases, the internal phase volume of PAM porous material also increases accordingly, making it able to encapsulate more cetyl alcohol, but at the same time, more and more internal phase makes the wall of the porous material gradually thinner, and through holes are more likely to appear between the holes and cause leakage of cetyl alcohol, which in turn leads to a decrease in the value of *R*. Therefore, the phase change enthalpy and the value of encapsulation rate were the highest in PCMs-3. The corresponding calculated encapsulation amount of the composite phase materials is depicted in Figure 2b. As seen, the maximum encapsulation amount, 6.13 g, is close to the adding value, demonstrating that polyacrylamide is a good coating carrier for cetyl alcohol and sodium sulfate decahydrate/diatomite.

The effect of emulsifiers, Tween-85, with different mass fractions on the properties of organic-inorganic coupling phase change materials was further tested and analyzed. Five sets of samples were prepared by adding the emulsifier of Tween-85 with different mass fractions and fixing the cetyl alcohol mass in the phase change material of 6.000 g, keeping an unchanged mass of sodium sulfate decahydrate/diatomite composite material of 0.300 g. A differential scanning calorimeter was used to characterize these five groups of samples and the DSC curves obtained by scanning are displayed in Figure 3a. From the DSC curve, the phase change latent heat of the organic-inorganic coupled phase change material prepared by the different emulsifier, Tween-85, mass fraction is calculated by the area of the curve, which is integrated using the software of the differential scanning calorimeter. The phase change latent heat data are shown in Figure 3b. It can be seen from the graph that when the added amount of emulsifier, Tween-85, in the experiment is 3.38%, the phase change latent heat of the coupled phase change material is up to 157.50 J/g. With the continuous increase in the amount of emulsifier Tween 85, the latent heat of phase change of the organic-inorganic coupled phase change material first increased, then decreased, and finally approached a steady trend. The occurrence of this situation indicates that the degree of emulsification of the oil-in-water emulsion in the emulsion polymerization gradually increases with the continuous addition of the oil-in-water emulsifier of Tween 85. The amount of the coated phase change material becomes smaller with the smaller size of the oil-in-water emulsion, and finally it reaches a plateau.

The single factor method was used to test and analyze the influence of different mass fractions of sodium sulfate decahydrate/diatomite on the thermal storage performance of the coupled phase change material. Multiple sets of parallel experiments were carried out by changing the added mass fraction of sodium sulfate decahydrate/diatomite composite material. A differential scanning calorimeter was used to characterize the coupled materials, and the DSC curves obtained by scanning are shown in Figure 4a. The area integration of the DSC curve in Figure 4a is performed to obtain the latent heat of phase change of the organic-inorganic coupled phase change material prepared by adding different mass fractions of sodium sulfate decahydrate/diatomite in the composite material. The drawn graph from the data is exhibited in Figure 4b. It can be seen from the figure that, the optimal group with the highest phase change latent heat of the phase change material is 3.40% addition amount of sodium sulfate decahydrate/diatomite in composite material, and the latent heat of phase change is 154.60 J/g in this experiment. With the increase of the mass fraction of sodium sulfate decahydrate/diatomite in the composite material, the latent heat of the organic-inorganic coupled phase change material shows a trend of first increasing and then decreasing. Within a certain range, as the mass fraction of the inorganic phase change material sodium sulfate decahydrate increases, the amount of the inorganic phase change material coated in the oil-in-water emulsion also increases. After reaching the coated threshold, as the amount of the inorganic phase change material increases, the inorganic phase change material is no longer effectively coated, resulting in the coating rate no longer increasing. The appearance of the uncoated sodium sulfate decahydrate material causes the demulsification of the oil-in-water emulsion. The oil phase in the oil-in-water emulsion is no longer completely coated, and the coating rate of the phase change material no longer increases, showing a decreasing trend.

### 3.2. Performance Characterization of Oic-PCM

The organic-inorganic coupling phase change material was prepared under the above optimal conditions. The total mass of the sample was 6.600 g, in which the organic phase change material cetyl alcohol: emulsifier (Tween 85): sodium sulfate decahydrate/diatomite composite material = 20:1:1. DSC curves were obtained by using the differential scanning calorimeter as shown in Figure 5a. It can be seen from the figure that the phase change intervals of the organic-inorganic coupling phase change material is between 30.16 °C and 56.03 °C. The specific phase transition temperature is 50.33 °C, the latent heat of phase transition is 164.13 J/g, and the coating ratio is 67.58%. In order to determine whether the phase change material has a good coating effect, a differential scanning calorimeter was used on the obtained organic-inorganic coupling phase change material to perform temperature rise and fall cycles test with a temperature rate of 5 °C. The variation of the corresponding phase transition enthalpy after 50 thermal cycles is displayed in Figure 5b. It can be seen that the phase change peak value of the material does not change significantly after 50 cycles of heating and cooling and the retention rate is as high as 97.59%. The decrease of phase change enthalpy is mainly due to the loss of bound water in Na_2_SO_4_·10H_2_O and slight cetyl alcohol leakage. The enthalpy of composite PCM was still reasonable after 50 thermal cycles. Therefore, composite PCM showed good thermal reliability after thermal cycles. It also demonstrated that the organic-inorganic coupling phase change material prepared in this experiment has a good coating effect without obvious leakage. The reason for this phenomenon is that the addition of porous diatomite effectively reduces the flow of organic phase in the PCMs and prevents the leakage of cetyl alcohol. At the same time, this illustrates that the inorganic phase change material, sodium sulfate decahydrate, has no acute phase separation phenomenon in the organic-inorganic coupling phase change material. It is proven that diatomite can effectively adsorb the hydrated inorganic salt in its porous structure. The addition of diatomite is beneficial to the circulation of coupled phase change materials. Further, the thermal conductivity was measured on a thermal conductivity tester by weighing a certain amount of organic-inorganic coupling phase change material and organic phase change material of cetyl alcohol. The thermal conductivity of cetyl alcohol is 0.1999 W/(m·k) while that of organic-inorganic coupling phase change material is 0.2061 W/(m·k). Comparing the results, it is easy to find that the thermal conductivity of the organic-inorganic coupled phase change material has increased. Moreover, the material combinations and corresponding phase transition temperatures and endothermic enthalpies are summarized in Table 1. It is shown that the addition of diatomite and sodium sulfate decahydrate is beneficial to the heat conduction of the coupled phase change material.

The appearance and morphology of the emulsion microspheres were observed by scanning electron microscope. The results are shown in Figure 6. The prepared emulsion microspheres are uniformly dispersed, form a good spherical structure, and have a dense surface. This structure can effectively prevent the leakage of liquid materials to increase the service life. The particle size analysis shows that the particle size distribution is concentrated between 42 μm and 49 μm, indicating that the preparation method is reliable and the product has good expected performance.

The organic-inorganic coupling phase change material, cetyl alcohol, diatomite, sodium sulfate decahydrate, sodium sulfate decahydrate/diatomite composite material, and polyacrylamide were scanned using an X-ray diffractometer and the scanning results are shown in Figure 7a and b. It can be seen from the figure that the XRD pattern of diatomite has a characteristic peak at 2θ = 20.45°. The XRD pattern of sodium sulfate decahydrate showed characteristic peaks at 2θ = 28.75°, 30.14°, and 33.73°. The XRD pattern of sodium sulfate decahydrate/diatomite composite material showed characteristic peaks of diatomaceous earth and sodium sulfate decahydrate in 2θ = 20.45°, 28.75°, 30.14°, and 33.73°, proving that sodium sulfate decahydrate can be effectively adsorbed in the porous structure of diatomite. There is no new characteristic peak in the XRD pattern of the sodium sulfate decahydrate/diatomite composite material, which proves that the diatomite and sodium sulfate decahydrate act together through physical methods, and there is no chemical reaction to generate new substances. It can also be seen from Figure 7b that the XRD pattern of cetyl alcohol has characteristic peaks at 2θ = 21.14° and 24.87°. The XRD pattern of the coupled phase change material also showed characteristic peaks in the corresponding positions. All characteristic peaks in cetyl alcohol can be found in the XRD patterns of organic-inorganic coupling phase change materials, indicating that polyacrylamide can effectively coat cetyl alcohol, and the two are combined through physical action. The characteristic peaks in the XRD patterns of cetyl alcohol, sodium sulfate decahydrate, and diatomaceous earth can all correspond to the peaks in the XRD patterns of the coupled phase change material. No new chemical substances were generated during this experiment, which confirmed that the composite phase change material was successfully prepared.

The thermal stability of the organic-inorganic coupling phase change material was researched by weighing a certain amount of organic-inorganic coupling phase change material and using a thermogravimetric analyzer. The measured TG curve is shown in Figure 8. It can be seen from the figure that the temperature at which the organic phase change material cetyl alcohol begins to decompose in the thermogravimetric analysis chart is 136.05 °C, and the temperature at which it completely decomposes is 248.05 °C. Meanwhile, the decomposition of organic-inorganic coupled phase change materials is divided into two stages: the temperature in the first stage is 28.48~246.71 °C and the temperature in the second stage is 246.71~496.40 °C. The mass loss in the first stage is the water content in the sodium sulfate decahydrate/diatomite composite material and the part mass of the organic phase change material, that is cetyl alcohol. The mass lost in the second stage is derived from the polyacrylamide shell material in the coupled organic-inorganic composite phase change material. Analysis of the TG curve shows that the organic-inorganic coupling phase change material prepared in the experiment has better thermal stability below 50 °C.

In order to detect the leakage, three parts of 1.000 g organic-inorganic coupling phase change materials were weighed and put in an oven at 45–65 °C. Sampling was performed every 10 °C, and each sample was dried for 60 min. The material was taken out and the leakage was observed. The results are shown in Figure 9 and the temperature from left to right is 45 °C, 55 °C, and 65 °C. It can be seen from the figure that the organic-inorganic coupled phase change material has no obvious change after heating at 45 °C for 60 min, and the retention rate reaches 95.6%, showing good thermal stability. After heating at 55 °C for 60 min, there was a slight leakage, and the retention rate was 85.3%. After heating at 65 °C for 60 min, it was found that the material leaked seriously, and the retention rate was only 60.3%. The application environment of the material is room temperature, that is, 25~40 °C. The experiment proves that, in this temperature range, the material has high phase change heat storage and no leakage.

## 4. Conclusions

In this work, an organic-inorganic coupled phase change material, with polyacrylamide as the wall material, organic phase change material cetyl alcohol, and sodium sulfate decahydrate adsorbed by diatomaceous earth with porous structure as the core material, was prepared via emulsion polymerization and testing was carried out. When the mass fraction of cetyl alcohol of the organic phase change material is 68.97%, the mass fraction of Tween-85 is 3.38%, and the mass fraction of sodium sulfate decahydrate/diatomite is 3.40%, the phase change latent heat of the coupled phase change material is the highest, and can reach as high as 164.13 J/g. Moreover, its coating rate is up to 67.58%. Meanwhile, the prepared material has good thermal storage cycle performance. After 50 thermal cycles, the phase change enthalpy only drops by 2.41%. The XRD pattern of the coupled phase change material contains the characteristic peaks of cetyl alcohol, diatomaceous earth, and sodium sulfate decahydrate. It is proven that, in the experiment, polyacrylamide coating cetyl alcohol is a physical change, and the adsorption of sodium sulfate decahydrate by diatomaceous earth also occurs through physical action. The test results of the thermal conductivity of the material show that the addition of diatomite and sodium sulfate decahydrate improves the thermal conductivity of the organic-inorganic coupled phase change material. Thermogravimetric analysis found that the organic-inorganic coupling phase change material has good thermal stability below 50 °C. The organic-inorganic coupling phase change material prepared in this experiment has no leakage in an environment of 45 °C. In the environment at 55 °C and 65 °C, there is some leakage. Combined with the analysis of the phase transition temperature of the material, it is concluded that the use environment of this material is 30~50 °C, which meets the actual application requirements. Although the emulsion particles prepared in this study achieved the effective combination of organic-inorganic composite materials and significantly reduced the leakage of the liquid phase, there was still a problem in that the coating rate could not be significantly improved. Future work still needs to consider different organic-inorganic material combinations to achieve better results.

## Data Availability

Not applicable.

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
