# Peer review of "Preparation of Organic-Inorganic Coupling Phase Change Materials with Enhanced Thermal Storage Performance via Emulsion Polymerization"

_materials, 2022, doi:10.3390/ma15093373_

Round 1

Reviewer 1 Report

1.The objective of the study is not clearly stated in the introduction section. Overall the introduction section is weak. Kindly do more literature survey and clearly mention the research gap.

2. DSC results must be presented in table form.

3. SEM analysis must be conducted to understand morphology of the prepared composite.

4. The leakage test must be supported by weight loss data of the samples.

5. Overall language used in the article is of below standard. There are numerous grammatical error through out the manuscript.

6. Kindly provide data of samples preparation in table form in weight 5 and in gram.

7. In conclusion section kindly add limitations of this study also.

Author Response

1.The objective of the study is not clearly stated in the introduction section. Overall the introduction section is weak. Kindly do more literature survey and clearly mention the research gap.

Response: Thank you for your valuable advice. Based on the previous revision, we have rewritten the section of introduction with reference to your suggestions and relevant literature. In the revised manuscript (modified sections are marked in red), based on literature research, we clearly articulate the research gaps.

  1. DSC results must be presented in table form.

Response: According the suggestion from reviewer, we added Table 1 to present the data from DSC results with relevant notes and discussions in revised manuscript.

Table. 1 Material combination and corresponding phase transition temperature and endothermic enthalpy.

Emulsifier %

cetyl alcohol

%

SSD/diatomite

%

Tm

ΔH

J/g

The coating rate

%

2.86%

73.98%

2.86%

50.48

135.12

55.41%

3.12%

71.62%

3.12%

50.45

153.86

63.22%

3.41%

68.97%

3.41%

50.33

164.13

67.58%

3.81%

65.32%

3.81%

49.74

153.71

63.41%

4.36%

60.33%

4.36%

50.48

130.20

53.86%

5.45%

67.30%

3.36%

49.27

142.03

58.48%

4.47%

67.99%

3.40%

49.74

146.40

60.28%

3.38%

68.77%

3.44%

49.88

157.50

64.85%

2.28%

69.55%

3.48%

49.95

148.77

61.26%

1.17%

70.34%

3.52%

49.45

140.69

57.93%

3.39%

67.85%

4.67%

49.51

118.36

48.73%

3.42%

68.31%

4.02%

49.31

140.72

57.94%

3.44%

68.75%

3.40%

50.22

154.60

63.66%

3.46%

69.22%

2.74%

49.27

135.93

55.97%

3.49%

69.73%

2.03%

49.29

116.61

48.01%

  1. SEM analysis must be conducted to understand morphology of the prepared composite.

Response: Thank you for your valuable and thoughtful comments. As you suggested that the SEM analysis should be conducted to understand morphology of the prepared composite. In revised version, we offered SEM images in Fig 6. The corresponding discussions are also added. As it is stated as “The appearance and morphology of the emulsion microspheres were observed by scanning electron microscope. The results are shown in Fig 6. The prepared emulsion microspheres are uniformly dispersed, form a good spherical structure and have a dense surface. This structure can effectively prevent the leakage of liquid materials. to increase service life. The particle size analysis shows that the particle size distribution is concentrated between 42 μm and 49 μm, indicating that the preparation method is reliable and the product has good expected performance.”

Fig.6 (a)、(b)SEM micrographs of the emulsion microspheres (c) particle size distribution.

  1. The leakage test must be supported by weight loss data of the samples.

Response: Thank you for your instructive suggestions. As the suggestion, we have further supplemented the leaked experimental data and graph of Fig.9 in revised manuscript. Corresponding discussions are also made. It can be seen from the figure that the organic-inorganic coupled phase change material has no obvious change after heating at 45℃ for 60 min, and the retention rate reaches 95.6%, showing good thermal stability. After heating at 55℃ for 60 minutes, there was a slight leakage, and the retention rate was 85.3%. After heating at 65℃ for 60 minutes, it was found that the material leaked seriously, and the retention rate was only 60.3%. The application environment of the material is room temperature, that is, 25~40 ℃. The experiment proves that in this temperature range, the material has high phase change heat storage and no leakage.

Fig. 9 Leakage detection analysis of organic-inorganic coupled phase transformation materials(oic-PCM) under different heating temperature of (a) 45℃, (b) 55℃ and (c) 65℃ after 60 min, (d) retention at different temperatures and times

  1. Overall language used in the article is of below standard. There are numerous grammatical error through out the manuscript.

Response:Thank you very much to point out the sentence structure and grammatical issues in our manuscript. According to the comments from you and the editors, we polished the manuscript with a professional assistance in writing, conscientiously. As in revised manuscript, changes are marked in red.

  1. Kindly provide data of samples preparation in table form in weight and in gram.

Response: According to your helpful advice, in revised version, we use Scheme 1to show the preparation process for organic-inorganic coupling phase change materials and data of samples preparation are labeled in table 1.

Scheme 1 Diagram of the preparation process for organic-inorganic coupling phase change materials.

Table. 1 Material combination and corresponding phase transition temperature and endothermic enthalpy.

Emulsifier %

cetyl alcohol

%

SSD/diatomite

%

Tm

ΔH

J/g

The coating rate

%

2.86%

73.98%

2.86%

50.48

135.12

55.41%

3.12%

71.62%

3.12%

50.45

153.86

63.22%

3.41%

68.97%

3.41%

50.33

164.13

67.58%

3.81%

65.32%

3.81%

49.74

153.71

63.41%

4.36%

60.33%

4.36%

50.48

130.20

53.86%

5.45%

67.30%

3.36%

49.27

142.03

58.48%

4.47%

67.99%

3.40%

49.74

146.40

60.28%

3.38%

68.77%

3.44%

49.88

157.50

64.85%

2.28%

69.55%

3.48%

49.95

148.77

61.26%

1.17%

70.34%

3.52%

49.45

140.69

57.93%

3.39%

67.85%

4.67%

49.51

118.36

48.73%

3.42%

68.31%

4.02%

49.31

140.72

57.94%

3.44%

68.75%

3.40%

50.22

154.60

63.66%

3.46%

69.22%

2.74%

49.27

135.93

55.97%

3.49%

69.73%

2.03%

49.29

116.61

48.01%

  1. In conclusion section kindly add limitations of this study also.

Response: Thank you for your valuable advice. Based on the previous revision, we further address the limitations of this study. It is stated in revised manuscript as follow:

 Although the emulsion particles prepared in this study achieved the effective combination of organic-inorganic composite materials and significantly reduced the leakage of the liquid phase, there was still a problem that the coating rate could not be significantly improved. Future work still needs to try different organic-inorganic materials combination to achieve better results.

Reviewer 2 Report

The authors herein present an important and interesting study on the synthesis and application of organic-inorganic phase change materials. The reviewer recommends publication of this work after addressing the following points:

  1. The introduction of this article seems lengthy. The authors may consider shorten the length of the introduction section so that the readers can clearly get the critical points.
  2. Since the emulsion polymerization is an important part of the experimental design of this work, the authors should include a scheme clearly showing the synthesis of the desired materials. Include the chemical structures of the starting materials (monomers), emulsifier, reaction medium, and the desired polymer structure. This will greatly help readers clearly understand the discussion, such as section 3.1 and 3.2.
  3. One of the biggest selling point of this work is the "good thermal storage performance" of the material. However, the authors only showed the thermal storage performance of the material at 45 C for only 30 minutes and claim a good storage performance. To further verify this claim, the authors should test the thermal storage performance of the material over a longer period of time. This will make the paper stronger and give valuable information to the readers.

Author Response

Reviewer 2:

The authors herein present an important and interesting study on the synthesis and application of organic-inorganic phase change materials. The reviewer recommends publication of this work after addressing the following points:

1.The introduction of this article seems lengthy. The authors may consider shorten the length of the introduction section so that the readers can clearly get the critical points.

Response: Thank you very much to point out the structure issues of the introduction section in our manuscript. As your advice, we have shortened the preface and adjusted the structure and presentation in revised manuscript and marked with red.

  1. Since the emulsion polymerization is an important part of the experimental design of this work, the authors should include a scheme clearly showing the synthesis of the desired materials. Include the chemical structures of the starting materials (monomers), emulsifier, reaction medium, and the desired polymer structure. This will greatly help readers clearly understand the discussion, such as section 3.1 and 3.2.

Response: Thank you very much. As the suggestion from you, we add the scheme to show the preparation process for organic-inorganic coupling phase change materials including the chemical structures of the starting materials (monomers), emulsifier, reaction medium, and the desired polymer structure in Scheme 1.

Scheme 1 Diagram of the preparation process for organic-inorganic coupling phase change materials.

  1. One of the biggest selling point of this work is the "good thermal storage performance" of the material. However, the authors only showed the thermal storage performance of the material at 45 C for only 30 minutes and claim a good storage performance. To further verify this claim, the authors should test the thermal storage performance of the material over a longer period of time. This will make the paper stronger and give valuable information to the readers.

Response: Thank you for your instructive suggestions. As the suggestion, we have further supplemented the leaked experimental data over a longer period and graph of Fig.9 in revised manuscript. Corresponding discussions are also made. It can be seen from the figure that the organic-inorganic coupled phase change material has no obvious change after heating at 45℃ for 60 min, and the retention rate reaches 95.6%, showing good thermal stability. After heating at 55℃ for 60 minutes, there was a slight leakage, and the retention rate was 85.3%. After heating at 65℃ for 60 minutes, it was found that the material leaked seriously, and the retention rate was only 60.3%. The application environment of the material is room temperature, that is, 25~40 ℃. The experiment proves that in this temperature range, the material has high phase change heat storage and no leakage.

Fig. 9 Leakage detection analysis of organic-inorganic coupled phase transformation materials(oic-PCM) under different heating temperature of (a) 45℃, (b) 55℃ and (c) 65℃ after 60 min, (d) retention at different temperatures and times

Round 2

Reviewer 1 Report

The authors have updated the manuscript as per the reviewer comments. It can now be considered for publication.